# EXPANDING THE WEB, SMALLER IS BETTER: A COMPREHENSIVE STUDY IN POST-TRAINING

## ABSTRACT

General-purpose large language models (GLLMs) like GPT-4 and LLaMA have demonstrated exceptional performance across a wide range of tasks. However, their performance often falls short in domain- or task-specific applications, where deeper, specialized knowledge is essential, while maintaining general knowledge remains crucial for handling broader, unseen tasks. Post-training has been widely applied to make LLMs specialized, typically consisting of multiple stages, including Domain-Adaptive Pre-Training (DAPT) and Supervised Fine-Tuning (SFT). In this work, we conduct a comprehensive study on three key aspects of post-training taking Finance as a target domain: (1) the distinct roles of DAPT and SFT in post-training, (2) strategies to mitigate knowledge forgetting across stages, and (3) evaluation methods that capture both general and domain-specific capabilities.

Our results show that DAPT and SFT require distinct training objectives, joint training of DAPT and SFT is essential for maintaining stage knowledge and encouraging knowledge transfer across stages, and replay mechanisms are critical for preventing forgetting. Evaluation should encompass general, seen, and unseen tasks for a complete assessment. Based on these insights, we developed a Joint-and-Replay post-training recipe and built LLaMA3-8B-Fin, a smaller yet more powerful state-of-the-art financial LLM trained through post-training. Despite its smaller size, LLaMA3-8B-Fin surpasses larger models like GPT-4o and LLaMA3.1-70b on both seen and unseen financial tasks while retaining general knowledge, demonstrating that a well-structured post-training can "expand the web" of capabilities in smaller LLMs, enabling them to outperform much larger models.

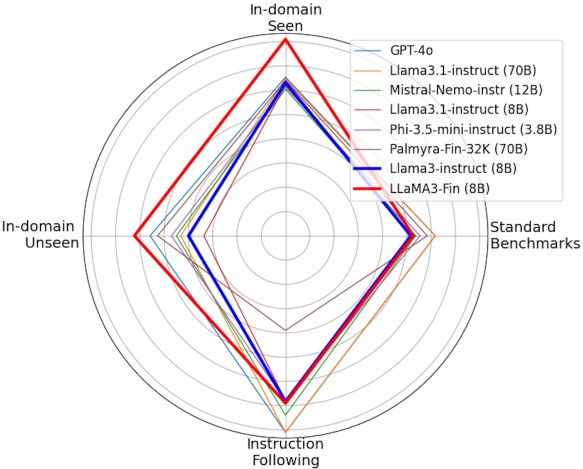

Figure 1: The model built with Joint-and-Replay post-training, LLaMA3-8B-Fin (red), "expands the web" of its base model, LLaMA3-8b-Instruct (blue), achieving better performance in finance-specific tasks (on both seen and unseen during SFT) while retaining general skills without forgetting (on both standard and instruction-following benchmarks). While it is smaller, it outperforms significantly larger models, such as GPT-4o and LLaMA3.1-70b-Instruct. More details can be found in Section 8.

# 1 INTRODUCTION

In recent years, we have witnessed the rise of General-purpose Large Language Models (GLLMs), such as GPT-4 (OpenAI, 2023), Claude (Anthropic, 2024), PaLM (Chowdhery et al., 2023), and LLaMA (Llama, 2024), to name a few. These models demonstrate impressive capabilities across a wide range of tasks. However, when it comes to real-world applications, these GLLMs often fall short. Many critical use cases often require domain-expert LLMs, such as those used in legal (Colombo et al., 2024), medical (Chen et al., 2023), or financial (Li et al., 2023) contexts. Task-specific LLMs, fine-tuned for particular objectives like code generation (Roziere et al., 2023) or retrieval-augmented generation (Nguyen et al., 2024), and personalized LLMs (Salemi & Zamani, 2024) that customize interactions tailored to individual users, demand models that go beyond generalization and are optimized for specific domains. Moreover, knowledge is constantly evolving, and a pre-trained LLM can become outdated shortly after deployment. Continuously injecting up-to-date, specialized knowledge into these models is crucial. Additionally, as GLLMs scale in size, their computational overhead becomes prohibitively high, making them impractical and costly to deploy at scale. Therefore, specialized LLMs not only offer better performance but also provide a more efficient, scalable solution for addressing complex, dynamic challenges.

To develop an effective domain specialized LLM, two primary goals must be met: (1) injecting deeper, domain-specific knowledge to enhance expertise in specialized tasks, and (2) maintaining strong general-purpose capabilities. This is crucial because domain-specific data is usually insufficient to cover general knowledge, leading to difficulties when end tasks require a combination of specialized and general knowledge (e.g., tasks not seen during supervised fine-tuning). We call the training process to achieve these goals as **post-training**. Starting from a GLLM, post-training involves additional training with above goals in mind. Although prior work has explored various aspects of post-training, most approaches merely involve additional pre-training on specialized data (Xie et al., 2023a), or rely on the traditional LLM framework where a single pre-training stage is followed by task-specific fine-tuning via classifiers (Ke et al., 2023). Some approaches simply regard post-training the same as continual learning, without considering the stage dependencies that are unique to modern LLM pre-training or the restrictions on access to pre-training data (Colombo et al., 2024). These approaches are insufficient to meet the increasing complexity of today's LLM applications.

In this paper, we focus on GLLMs that undergo *multi-stage* training (i.e., instruction-tuned GLLMs) and *multi-stage* post-training, which includes Domain-Adaptive Pre-Training (DAPT) and domain-specific Supervised Fine-Tuning (SFT). DAPT aims to learn the background knowledge from raw text, while SFT focuses on instruction learning. Instead of naively training sequentially with specialized data, we investigate critical research questions, including (1) *distinct roles of DAPT and SFT*, (2) *approaches for mitigating forgetting*, and (3) *effective evaluation methods for post-training systems*. To explore these, we conduct targeted experiments. For (1), we perform ablations on training pipeline, examining sequential versus joint training and the impact of different loss functions across stages. For (2), we investigate replay-based techniques (Rebuffi et al., 2017), such as mixing general and domain-specific data, and modular-based approaches like parameter-efficient fine-tuning or PEFT (Hu et al., 2021) to mitigate forgetting. For (3), we explore the evaluation strategies that can evaluate both the domain-specific knowledge and general knowledge in the LLM.

| RQ1: What is the role of DAPT and SFT in post-training? |
|---|
| - DAPT uses next-token prediction, while SFT needs instruction masking added. §5.1 |
| - Both DAPT and SFT contribute to improvements. §5.2 |
| - Joint training with DAPT and SFT yields better results than sequential training. §5.3 |
| **RQ2: How to mitigate forgetting in post-training?** |
| - Two types of forgetting phenomena observed between GLLM and post-training: general knowledge and stage-specific knowledge (e.g., the instruction-following knowledge from SFT stage). §6.1 |
| - Negligible forgetting observed within the post-training stage. §6.1 |
| - Replay-based approaches are most effective, especially with a mix of general, in-domain, DAPT, and SFT data. §6.2 |
| - Modular approaches like PEFT help prevent forgetting but are less effective than full model fine-tuning. §6.2 |
| **RQ3: How to evaluate post-training?** |
| - Evaluate general capabilities using standard and instruction-following benchmarks. §7 |
| - Evaluate in-domain performance using seen and unseen tasks ("seen" refers to task types covered during SFT). §7 |

Table 1: Summary of research questions and corresponding key findings.

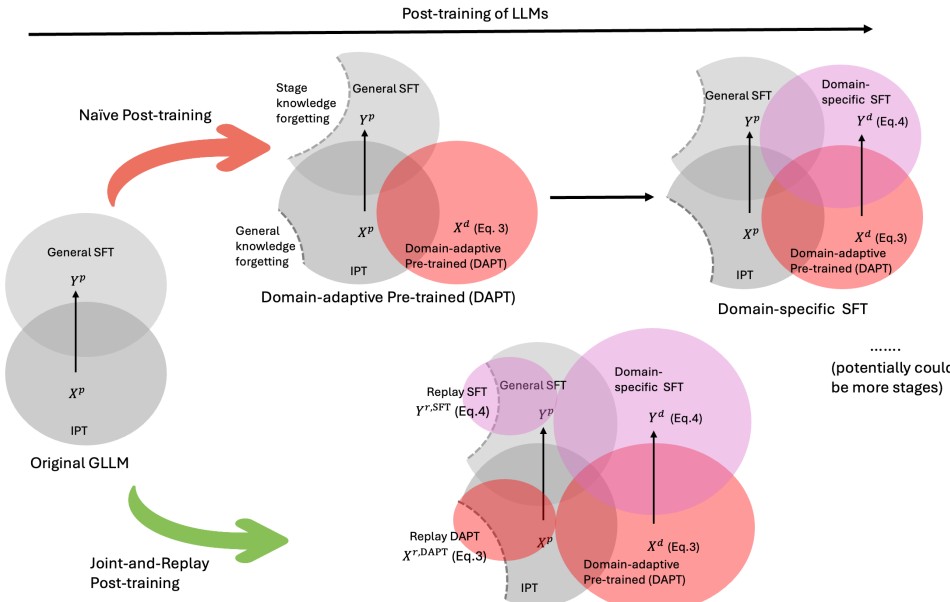

Figure 2: Conceptual overview of naive post-training and Joint-and-Replay post-training. Circles indicate the knowledge learned on each stage. Grey color refer to the stages that are outside of post-training. Red and purple colors indicate DAPT and SFT stages in post-training respectively. The original GLLM undergoes multiple stages, here we consider only initial pre-training (IPT) and SFT. The uncolored segment indicates the amount of forgetting.

Table 1 summarizes our key findings with respect to the research questions. Notably, we find that DAPT and SFT play *complementary roles* in enhancing post-training performance, with joint training of the two stages yielding better results than sequential training. We also identified two types of forgetting: the first involving *general knowledge*, and the second concerning *stage-specific knowledge* (e.g., instruction-following skills learned during SFT). To mitigate these forgettings, *replay*, which involves mixing additional general data with domain-specific data shown to be particularly effective.

Building on these insights, we propose a new training recipe, **Joint-and-Replay Post-training**, as shown in Fig. 2. Starting from a GLLM, naive post-training introduces two additional sequential stages. The first is DAPT, aimed at extending the pre-trained knowledge with domain-specific knowledge, but this may lead to forgetting general and stage-specific knowledge (missing pieces are shown in the grey circles). Similarly, domain-specific SFT broadens task learning within the domain but may further forget general task knowledge, though some forgotten stage knowledge might be re-learned in this stage (larger missing piece in initial pre-training (IPT) while smaller missing piece in general SFT). To overcome these challenges, Joint-and-Replay post-training jointly trains DAPT and SFT with appropriate mixture ratios to mitigate stage-specific knowledge. It also mixes the domain-specific data with general data to mitigate general knowledge forgetting. For loss computation, it masks the instruction part in the SFT data. To demonstrate the effectiveness of Joint-and-Replay post-training, we conduct post-training on the popular financial domain as a case study, resulting a new financial LLM, **LLaMA3-8B-Fin**. Extensive experiments show that LLaMA3-8B-Fin is a new state-of-the-art LLM in financial domain. Despite its smaller size, it outperforms much larger models, such as GPT-4o and LLaMA3.1-70b, on both seen and unseen finance-related tasks, while also showing no degradation on general benchmark tasks (Fig. 1).

In summary, our key contributions include:

- To our knowledge, this is the first comprehensive analysis of post-training using contemporary LLMs, addressing key research questions and identifying critical factors that influence post-training's effectiveness.

- Based on insights from above analysis, we propose Joint-and-Replay post-training, an effective training recipe that incorporates replay of both stage-specific and general knowledge data, along

with joint training of post-training stages. Additionally, we present a comprehensive evaluation protocol that accounts for both general and domain-specific capacities for post-training.

- To demonstrate the effectiveness of our training recipe, we built LLaMA3-8B-Fin, a state-of-the-art financial LLM that, despite its smaller size, outperforms much larger models in the financial domain and exhibits no forgetting of general capacities. This highlights that, with our training recipe, expanding the knowledge of an LLM is highly achievable, and a smaller LLM can be better than significantly larger models.

## 2 RELATED WORK

**Continual learning and catastrophic forgetting.** Post-training is closely related to continual learning, which focuses on learning a sequence of tasks sequentially without forgetting (Chen & Liu, 2018; McCloskey & Cohen, 1989; Van de Ven & Tolias, 2019; Mai et al., 2022; Aljundi et al., 2019). Typical approaches include *regularization-based methods* that regularize parameter updates to preserve important parameters (Kirkpatrick et al., 2016; Seff et al., 2017); *modular-based methods* that dynamically modify the architecture (Serrà et al., 2018; Wortsman et al., 2020); and *replay-based method* that recall previous experiences (Rebuffi et al., 2017; Wang et al., 2020). One might mistakenly view post-training as a form of continual learning with just two tasks—one being pre-training and the other post-training. However, there are significant differences between the two. First, foundational pre-training and post-training should not be simply considered as two tasks in a sequence, as they consist of multiple stages. post-training aims to preserve both general pre-trained knowledge and stage-specific knowledge within a GLLM, rather than focusing on task-specific knowledge (Lopez-Paz & Ranzato, 2017; Wortsman et al., 2020; Shin et al., 2017; Serrà et al., 2018; Zeng et al., 2019; Rebuffi et al., 2017). Second, unlike continual learning, post-training often cannot access the original pre-training data, making it impossible to compute the statistics that continual learning typically relies on (Varshney et al., 2022; Huang et al., 2021; Shen et al., 2019; Liu et al., 2019; de Masson d'Autume et al., 2019; Wang et al., 2020; Li et al., 2022; Wang et al., 2022b;a). Third, while the tasks in continual learning are typically independent or loosely related (Wang et al., 2021; Zhao et al., 2022; Jin et al., 2021), post-training involves strong task dependencies. The tasks progress from GLLM, to DAPT, and then to SFT, becoming increasingly aligned with the final task as the stages advance. These distinctions make continual learning methods unsuitable for direct application to post-training.

**Post-training.** Post-training has been widely adapted to GLLM to board domains, such as code (Nijkamp et al., 2022), medical (Luo et al., 2023), law (Colombo et al., 2024), mathematics (Azerbayev et al., 2023), multi-lingual (Chen et al., 2024) and finance (Xie et al., 2023a; Writer, 2024) and tasks such as function calling (Zhang et al., 2024), retrieval augmented generation (Nguyen et al., 2024; Ke et al., 2024) and LLM-as-a-judge (Wang et al., 2024). While many domain-specific or task-specific LLMs have been developed, with most following the standard post-training process (often including SFT, and optionally DAPT and RLHF). Some focus on domain-specific or task-specific data curation (Yang et al., 2024), auxiliary tasks (Wang et al., 2024), mixture ratio (Que et al., 2024), data-efficiency (Xie et al., 2023b) or hyper-parameters (Parmar et al., 2024). However, none have extensively investigated what constitutes an effective training recipe. Recently, Jiang et al. (2024) proposed "pre-instruction-tuning", where documents and QA pairs are trained together, similar to our joint DAPT and SFT training. However, their focus is primarily on QA tasks, and they do not evaluate general capabilities. In this work, we not only propose a post-training recipe that achieves state-of-the-art performance for financial LLMs, but more importantly, we also conduct a thorough investigation into various research questions related to post-training.

## 3 PROBLEM SETUP

Consider a GLLM that has undergone pre-training across $F$ stages, typically Initial Pre-Training (IPT), Supervised Fine-Tuning (SFT), and Preference Learning. We represent the multi-stage pre-training as:

$$\theta_i := \arg\min_{\theta} \mathcal{L}_i(\theta, D_i | \theta_{<i}), \tag{1}$$

where $L_i$ denotes the loss at stage $i \in \{1, 2, \ldots, F\}$, $D_i$ represents the training data for that stage, and $\theta_{<i}$ captures the model parameters trained in all previous stages. Post-training further trains the

LLM on top of the final pre-trained model $\theta_F$. Similar to pre-training, post-training can also consist of $C$ additional stages, typically mirroring the structure of the pre-training stages:

$$\theta_{F+j} := \arg\min_{\theta} \mathcal{L}_{F+j}(\theta, D_{F+j} | \theta_{<F+j}), \tag{2}$$

where $j \in \{1, 2, \ldots, C\}$. During post-training, it is a practical assumption that the model does not have access to the original pre-training data. In some cases, proxy data may be available, but this lack of access is a key factor contributing to pre-trained knowledge forgetting. Since pre-training consists of multiple stages, each with a distinct focus, post-training stages may cause forgetting of knowledge specific to earlier pre-training stages. For instance, Domain Adaptive Pre-Training (DAPT) during post-training may lead to forgetting knowledge from the SFT stage of pre-training, such as the model's instruction-following abilities.

## 3.1 STAGES IN POST-TRAINING

As shown in Eq. 2, post-training can include multiple stages. In this work, we focus on the two most common stages: DAPT and SFT.

**Domain-adaptive Pre-training (DAPT).** The goal of DAPT stage is to learn domain-specific background knowledge so that later stages can leverage. Typically, unsupervised data (raw text) is used in this stage and the training uses the next token prediction loss:

$$\mathcal{L}_{\text{DAPT}} = -\sum_{t=1}^{T_x} \log P_\theta(x_t | x_{<t}) - \lambda_{\text{replay}} \cdot \sum_{t=1}^{T_r} \log P_\theta(x_t^r | x_{<t}^r). \tag{3}$$

Here, $x_t$ and $x_t^r$ indicate the token at position $t$ in the in-domain and replay input sequences, respectively. $T_x$ and $T_r$ indicate the total number of tokens in an example from the domain-specific data and replay data, respectively. $\lambda_{\text{replay}}$ is a weighting factor for the replay loss, where "replay" refers to additional data that is mixed with domain-specific data to mitigate forgetting of general knowledge. Since we typically do not have access to the pre-training data, the replay data is usually *guessed* to be similar to the pre-training data, such as general domain data from Wikipedia. If we do not apply any replay ($\lambda_{\text{replay}} = 0$), the Eq. 3 is reduced to simple next token prediction on the domain-specific data.

**Supervised Fine-tuning (SFT).** Another important post-training stage is SFT, which is aimed to gain domain-specific instruction following ability. While it is generally agreed that SFT data is supervised consisting of instruction and answer (or user turn and asssitant turn if in chat format), there is no agreement on the optimal training loss[1]. There is, though, a growing agreement that the instruction part should be masked out during training. To investigate the impact of masking versus non-masking, we formulate a general form as:

$$\mathcal{L}_{\text{SFT}} = -\sum_{t=1}^{T_x} M_t \log P_\theta(x_t | x_{<t}) - \lambda_{\text{replay}} \cdot \sum_{t=1}^{T_r} M_t \log P_\theta(x_t^r | x_{<t}^r), \tag{4}$$

where $M_t$ indicates the *token mask*: $M_t = 1$ means the token is included in the loss and $M_t = 0$ indicates it is masked out. If $M_t = 1, \forall t$, then Eq. 4 is reduced to Eq. 3. If $M_t = 0$ for the instruction part, Eq. 4 becomes an *instruction mask loss*, i.e., excludes the instruction (or user term) in the loss computation.

## 4 EXPERIMENTAL SETUP

### 4.1 DATASETS TO STUDY POST-TRAINING

To explore the three research questions in Table 1, it is essential to prepare datasets that consist of DAPT, SFT, and the corresponding replay data. We use the popular financial domain as our case study. Specifically, we collected large-scale datasets and divided them into four sections: **DAPT-Gen** (general domain DAPT data), **DAPT-In** (in-domain DAPT data, specifically from the financial domain), and similarly, **SFT-Gen** and **SFT-In**. We further combined the general and in-domain

---

[1]To give an example, Ouyang et al. (2022); Zhou et al. (2024) use next token prediction, whereas Touvron et al. (2023) applies masking to the instruction part in the SFT loss.

| Stage | Type | Task | Datasets | Size |
|---|---|---|---|---|
| DAPT | General | Raw text | NaturalInstr PromptSource Math Aqua CREAK Esnli Qasc Soda Strategy-qa UnifiedSKG GSM8K ApexInstr DMMath Dialogstudio | 3.2B |
| | Finance | Raw text | Fineweb-fin | 3.7B |
| SFT | General | Math word | Orcamath | 200K |
| | | Math | Metamath | 395K |
| | | Instruction-follow | SelfInstruct | 82K |
| | | Augmented FLAN | Slimorca | 518K |
| | | Code instruction | MagicoderEvol | 111K |
| | | Conversation | Ultrachat | 208K |
| | | Conversation | Sharegpt | 90K |
| | | Math rationale | Mathinstruct | 262k |
| | Finance | Relation Classification | Finred | 27.6K |
| | | Entity Recognition | NER-cls | 13.5K |
| | | | NER | 511 |
| | | Headline Classification | Headline-cls | 82.2K |
| | | Sentiment Classification | Sentiment-cls | 47.6K |
| | | | Sentiment-train | 76.8K |

(a) DAPT and SFT data.

| Type | Task | Datasets | Size |
|---|---|---|---|
| General | Various | MMLU.... | — |
| Finance | Sentiment Classification | FPB | 970 |
| | | FiQA SA | 235 |
| | | FOMC | 496 |
| | Entity Recognition | NER | 98 |
| | ESG issue Classification | MLESG | 300 |
| | Rumour Detection | M&A | 500 |
| | Summarization | EDTSUM | 2K |
| | | ECTSUM | 495 |
| | QA Openformat | Finance Bench | 150 |
| | Stock Movement Predict | SM-Bigdata | 1.47K |
| | | SM-ACL | 3.72K |
| | | SM-CIKM | 1.14K |
| | Fraud Detection | CRA-CCF | 2.28K |
| | | CRA-CCFraud | 2.1K |
| | Credit Scoring | German | 200 |
| | | Astralian | 139 |
| | | LendingClub | 2.69K |
| | Distress Identidication | Polish | 1.74K |
| | | Taiwan | 1.37K |
| | Claim Analysis | ProtoSeguro | 2.38K |
| | | TravelInsurance | 2.53K |
| | Tabular QA | TATQA | 1.67K |

(b) Evaluation data. We use 11 standard benchmarks for general knowledge, including MMLU, AI2-ARC, PIQA, Social-IQA, GSM8K, MathQA, TriviaQA, Nq-open, Hellaswag, Winogrande, and Openbookqa.

Table 2: Training and evaluation datasets used in post-training. Dataset sizes in DAPT are measured in tokens, while SFT and evaluation sizes are based on the number of samples.

data to create **DAPT-Mix** and **SFT-Mix**[2]. Table 2 provides details on each section along with our evaluation set. For DAPT-In, we select finance-related data from FineWeb (Penedo et al., 2024) based on URLs. For DAPT-Gen, we curate a diverse range of tasks to ensure it represents the broad knowledge of a GLLM. The same strategy was applied to SFT-Gen. In the evaluation, only *two task types* (sentiment classification and named entity recognition) has been seen in the SFT data, and these are highlighted in grey. While it is possible that some general tasks *training data* (e.g., GSM8K) overlap with DAPT-Gen, we ensured that no evaluation data was seen during DAPT-Gen training. Unlike previous work (Luo et al., 2023; Colombo et al., 2024; Azerbayev et al., 2023; Xie et al., 2023a), which focused primarily on in-domain seen tasks, we also evaluated on general tasks and unseen in-domain tasks to provide a more comprehensive assessment of LLM performance.

### 4.2 POST-TRAINING AND EVALUATION

Our post-training starts from **LLaMA3-8-instruct**, and performs DAPT and SFT on top of this base model. We focus on *0-shot* performance (i.e., no in-context sample is given), as it directly reflects the effectiveness of the parametric knowledge embedded in the LLM. We use `llm-eval-harness`[3] to conduct the evaluation experiments. For general tasks, we employ the default setting in the package. For finance-specific tasks, we employ exact match for classification tasks (e.g., sentiment classification) and Rouge-1 score for generation tasks (e.g., summarization). All numbers reported are based on the average of three random seeds. The average results of all tasks within the corresponding section are reported and we left the detailed results of individual task to the Appendix B. The usage of chat-format and hyper-parameters can also be found in the Appendix A.

## 5 WHAT IS THE ROLES OF DAPT AND SFT IN POST-TRAINING?

### 5.1 SFT NEEDS INSTRUCTION MASKING ADDED

As discussed in Section 3.1, the primary difference between DAPT and SFT lies in the data: DAPT uses raw text, while SFT uses supervised task data. In modern LLMs, all tasks are unified in a generative format, with the model performing next-token prediction. Technically, DAPT and SFT

---

[2]No sampling is performed but simply combines the in-domain data and general domain data in Table 2. We leave investigating the optimal mixture ratio for future work.

[3]https://github.com/EleutherAI/lm-evaluation-harness

could use the same loss function. However, a common scenario is having significantly more DAPT data than SFT data, with the expectation that SFT should focus more on task-specific learning. We compared the performance of SFT with and without masking the instruction part in Table 3.

We can see that using instruction masking significantly improves performance on seen tasks but results in lower performance on unseen tasks, with the improvement on seen tasks being more pronounced. This suggests that SFT with instruction masking focuses heavily on task-specific knowledge, while without instruction masking, the model retains more general knowledge, which benefits unseen tasks. We also observed that for SFT without instruction masking, a chat-format is necessary; otherwise, the model tends to produce unreasonable outputs[4].

| Setting | General | Seen Finance | Unseen Finance |
|---|---|---|---|
| LLaMA3-8b-instruct | **51.65** | 63.16 | 39.94 |
| SFT (w/o instr mask) | — | 73.69 | **52.08** |
| SFT (instr mask) | 50.10 | **81.49** | 49.92 |

Table 3: Effectiveness of the insturction mask in SFT.

## 5.2 BOTH DAPT AND SFT CONTRIBUTE TO IMPROVEMENT IN SEQUENTIAL POST-TRAINING

While we focus on multi-stage post-training, there are work that focus on only one-stage approach (Cheng et al., 2024b;a; Xie et al., 2023a). Although a one-stage approach is appealing for mitigating forgetting across stages and can be more directly aligned with end-tasks (if known in advance), it may lose some transferable information across stages. We are particularly interested in the necessity of employing multi-stage post-training. Table 4 compares the base

| Setting | General | Seen Finance | Unseen Finance |
|---|---|---|---|
| LLaMA3-8b-instruct | 51.65 | 63.16 | 39.94 |
| SFT | 50.10 | **81.49** | 49.92 |
| DAPT → SFT | **54.84** | 81.05 | **56.50** |

Table 4: Effectiveness of SFT and DAPT. Results are average over all the tasks in the corresponding section as shown in Table 2.

(llama3-8b-instruct), SFT (with instruction mask), and DAPT → SFT (sequential training). The results show that while SFT improves over the base model, adding DAPT before SFT further outperforms SFT, especially on unseen tasks. This demonstrates that DAPT provides transferable background knowledge to SFT, enabling further improvements. We also observe improvement on general tasks. This is understandable as DAPT may contain the training data of some general tasks. For seen tasks, adding DAPT performs comparably to SFT, which is expected since seen tasks primarily benefit from the targeted improvements brought by SFT.

## 5.3 JOINT TRAINING WITH DAPT AND SFT IMPROVES

Since the introduction of Instruct-GPT (Ouyang et al., 2022), the 3-stage post-training process (Section 3) has become a widely accepted standard. However, we challenge this convention by evaluating both joint (DAPT + SFT) and sequential (DAPT → SFT) pipelines of DAPT and SFT. We also experimented with down-sampling DAPT to balance it with SFT, aiming to prevent distraction from excessive DAPT data. Table 5 shows the comparison. Sequential training significantly improves performance on unseen tasks compared to SFT alone, and joint training with down-sampling yields further improvements. This suggests that joint training facili-

| Setting | General | Seen Finance | Unseen Finance |
|---|---|---|---|
| LLaMA3-8b-instruct | 51.65 | 63.16 | 39.94 |
| SFT | 50.10 | **81.49** | 49.92 |
| DAPT(full) → SFT | **54.84** | 81.05 | 56.50 |
| DAPT(down) → SFT | 51.16 | 79.37 | 57.83 |
| DAPT (full) + SFT | 53.62 | 72.15 | 55.59 |
| DAPT (down) + SFT | 52.65 | 81.00 | **62.23** |

Table 5: Effectiveness of joint training and sequential training. "Full" indicates full data while "down" indicate data that is down-sampled. "→" indicates sequential training and "+" indicates joint training.

tates better knowledge transfer between DAPT and SFT. We can also see that DAPT (full) + SFT performs significantly worse on in-domain tasks compared to DAPT (down) + SFT. This suggests that too much DAPT data may have an adverse effect on performance, as it can distract from the

---

[4]Since our general tasks evaluation requires non-chat-format (as chat-format is too flexible for evaluations with fixed metrics), we could not report general task performance for SFT without instruction masking ("—" in the Table 3). We leave human and LLM-as-a-judge evaluations for future work.

primary goal (i.e., following instructions to solve tasks)[5]. We do note that DAPT (down) + SFT underperforms DAPT (full) → SFT. This is understandable as DAPT (full) contains more data, that could benefit general tasks. Additionally, SFT may dilute the general-task benefits gained from DAPT. However, DAPT (down) + SFT achieves the best overall balance.

# 6 HOW TO MITIGATE FORGETTING IN POST-TRAINING?

This section looks into the forgetting in post-training, i.e., the LLM should perform reasonably well on the learned skills. This can be measured by the performance drop in the general domain. Due to the computation limits, we use a subset of training and evaluation datasets to gain observations. Specifically, for SFT-Gen, we down-sampling it to be the same size as SFT-In, and evaluate only the first 8 tasks in Table 2. As a result, the results in this section are not directly comparable with Section 5.

## 6.1 FORGETTING IN POST-TRAINING

**General and stage knowledge forgetting between GLLM and post-training.** Since we do not have access to the original pre-training data, forgetting becomes an inevitable issue. Beyond general knowledge in GLLMs, we also need to address forgetting at different stages. For example, after DAPT in post-training, the model may forget how to follow instructions, as this capability is acquired during the SFT stage in pre-training. This stage mismatch can lead to stage-specific forgetting. In Table 6, we illustrate both types of forgetting. In the first section (rows 2-4), we apply only DAPT to an

| Setting | General | Seen Finance | Unseen Finance |
|---|---|---|---|
| LLaMA3-8b-instr | 51.65 | 63.16 | 47.63 |
| DAPT-In | 47.64 | 59.93 | 36.11 |
| DAPT-Gen | 54.03 | 49.42 | 45.62 |
| DAPT-Mix | 51.33 | 53.57 | 46.51 |
| DAPT-In → SFT-Mix | 50.30 | **69.91** | 42.06 |
| DAPT-Gen → SFT-Mix | **56.54** | 62.33 | **51.74** |
| DAPT-Mix → SFT-Mix | 55.25 | 68.85 | 51.04 |

Table 6: Effectiveness of joint training and sequential training.

instruction-tuned LLM, and observe a significant performance drop on finance tasks. This suggests that the model has forgotten how to solve the tasks, or more specifically, how to follow instructions, as the knowledge learned in the SFT stage has been lost. In the second section (rows 5-7), we apply SFT after DAPT, leading to improvements across all cases, as the model regains its instruction-following ability. We also observe that DAPT-In performs well on seen tasks but forgets general and unseen tasks, while DAPT-Gen excels on general and unseen tasks but performs worse on seen tasks. The best results are achieved with DAPT-Mix and SFT-Mix. These findings indicate that using only in-domain data causes forgetting of general knowledge, and replay is crucial to prevent such forgetting.

**Negligible forgetting within post-training stages.** We have observed stage-mismatch forgetting between GLLM and post-training. Given that post-training itself involves multiple stages, we are curious whether forgetting also occurs within post-training stages. Table 7 presents results when DAPT and SFT are mismatched (e.g., one with general domain data while the other with financial domain data). We observe that DAPT consistently improves the corresponding

| Setting | General | Seen Finance | Unseen Finance |
|---|---|---|---|
| LLaMA3-8b-instr | 51.65 | 63.16 | 47.63 |
| DAPT-In → SFT-Gen | 48.04 | **66.05** | 46.13 |
| DAPT-Gen → SFT-In | 54.15 | 58.42 | 42.36 |
| DAPT-Mix → SFT-In | 56.57 | 70.25 | **53.45** |
| DAPT-Mix → SFT-Gen | **57.48** | 61.75 | 48.94 |

Table 7: Negligible forgetting when DAPT and SFT are mimatched.

tasks (DAPT-In enhances performance on in-domain seen tasks, and DAPT-Gen improves general tasks), regardless of the SFT stage. This suggests that SFT does not induce forgetting of DAPT knowledge, as the stages within post-training is more transferable to one another.

## 6.2 MITIGATE THE FORGETTING

**Replay-based approach.** We have identified two types of forgetting that need to be addressed. As shown in Table 6, DAPT-Mix and SFT-Mix significantly reduce general knowledge forgetting. Furthermore, in Table 5, we observe that joint DAPT and SFT further improve performance, as there

---

[5]An interesting follow-up question is determining the optimal mixture ratio. We leave this for future work.

is no single isolated stage in post-training that can induce stage knowledge forgetting. These findings suggest that replaying data is both effective and essential for preventing forgetting in post-training.

**Modular-based approach.** Another popular method for preventing forgetting is the modular-based approach, which allocates specific parameters or models to particular tasks or domains. In our case, we use the widely adopted PEFT method, LoRA (Hu et al., 2021), with a rank size of 128. While LoRA has been shown to be effective for task-specific fine-tuning, we are interested in its utility within the post-training framework. In Table 8, we compare full fine-

| Setting | General | Seen Finance | Unseen Finance |
|---------|---------|--------------|----------------|
| LLaMA3-8b-instr | 51.65 | 63.16 | 47.63 |
| SFT (FT) | 51.20 | 68.31 | 41.31 |
| SFT (LoRA) | 50.57 | 68.55 | 42.36 |
| DAPT (FT) → SFT (LoRA) | 50.78 | 65.28 | 47.70 |
| DAPT (LoRA) → SFT (LoRA) | 50.92 | 66.20 | 50.72 |
| DAPT (FT) → SFT (FT) | **55.25** | **68.85** | **51.04** |

Table 8: Effectiveness of PEFT. "FT" indicates full fine-tuning.

tuning with LoRA for both SFT and DAPT+SFT. We observe that SFT (LoRA) performs similarly to SFT (FT), consistent with prior findings that LoRA is effective for task adaptation. We also find that DAPT (LoRA) further improves performance over SFT. However, these gains are still smaller compared to full model fine-tuning. This suggests that while PEFT is useful for both preventing forgetting and learning domain-specific knowledge, full fine-tuning yields even better results.

## 7 HOW TO EVALUATE POST-TRAINING?

As mentioned in the Introduction, post-training has two key objectives: to inject deeper, domain-specific knowledge into the LLM, and to preserve general knowledge so the model can effectively handle unseen tasks. This necessitates an evaluation framework that goes beyond in-domain seen tasks. Throughout our experiments, we divided our evaluation into two parts: **(1) general capacities** and **(2) in-domain capacities**. For (1), we included general standard benchmarks, and we will also include a general instruction-following benchmark in the following sections. These give us a more complete picture of the model's ability to prevent forgetting. For (2), we included both seen and unseen tasks, providing a comprehensive view of the model's performance across diverse tasks and its generalization ability.

## 8 JOINT-AND-REPLAY POST-TRAINING

Based on the insights from Sections 5-7, we develop Joint-and-Replay post-training, illustrated in Figure 2. Unlike naive post-training, which sequentially trains on domain-specific knowledge, we mix general and domain-specific data in both the DAPT and SFT stages to prevent forgetting of general knowledge ($\lambda_{\text{replay}} = 1$ in Eq. 3 and 4). To further mitigate stage-specific forgetting and encourage transfer between stages, we jointly train DAPT and SFT ($\mathcal{L}_{\text{Joint-and-Relay}} = \mathcal{L}_{\text{DAPT}} + \mathcal{L}_{\text{SFT}}$). Additionally, we down-sample the DAPT data to avoid distractions from an overemphasis on DAPT. DAPT employs next-token prediction, while SFT adds an instruction mask ($M_t = 0$ for the instruction part in $\mathcal{L}_{\text{SFT}}$ in Eq. 4). To extensively evaluate our model, besides those already in Section 4.2, we further evaluate our model on MT-bench (Zheng et al., 2023)[6], a popular benchmark to assess the general instruction-following ability.

We apply Joint-and-Replay post-training to post-train the LLaMA3-8b-instruction model on the financial domain, resulting in LLaMA3-8B-Fin. We compare this post-trained model against three different categories of baselines: **(1) general LLMs**, including GPT-4o (OpenAI, 2023), LLaMA3.1-70b-instruct (Llama, 2024), Mistral-Nemo-instruct (Jiang et al., 2023), LLaMA3.1-8b-instruct (Llama, 2024), and Phi-3.5-mini-instruct (Abdin et al., 2024), representing a range of sizes from 3.8B to large-scale models like GPT-4o; **(2) domain-specific LLM**, including the finance-specific Palmyra-Fin-32k (Writer, 2024), a recent state-of-the-art financial LLM[7]; **(3) post-training base model**,

---

[6]We use GPT-4 as the judge model and apply single-answer grading mode.

[7]We note that there are additional financial LLMs available, such as FinMa (Xie et al., 2023a) based on LLaMA2, Finance-LLM (Cheng et al., 2024b) based on LLaMA3-base and FinLLaVA (Xie et al., 2024) focuses on multi-modality and not publicly available. However, they are either significantly smaller in scale, based on less advanced LLMs compared to our model or not publicly available. In our preliminary experiments, these models performed considerably worse than both our model and the baselines. Therefore, we have only included the SoTA financial LLM in our comparisons.

LLaMA3-8b-instruct (Llama, 2024). This allows us to evaluate whether our post-training process can effectively "expand the web", i.e., enhance the base model's capabilities in the target domain while preserving its general skills and preventing forgetting.

| Category | Model | Size | General Capacities | | In-domain Capacities | |
| | | | Standard Benchmark | Instruction Following Benchmark | Seen | Unseen |
| --- | --- | --- | --- | --- | --- | --- |
| Domain-specific LLM | Palmyra-Fin-32K | 70B | 58.41 | 6.52 | 64.10 | 52.70 |
| General LLMs | GPT-4o | N/A | — | — | 65.41 | 55.79 |
| | LLaMA3.1-instruct | 70B | **61.70** | **9.14** | 64.68 | 43.17 |
| | Mistral-Nemo-instr | 12B | 53.58 | 8.70 | 60.42 | 44.96 |
| | LLaMA3.1-instruct | 8B | 51.89 | 8.33 | 62.01 | 33.61 |
| | Phi-3.5-mini-instruct | 3.8B | 55.43 | — | 61.31 | 47.01 |
| Post-train base model | LLaMA3-instruct | 8B | 51.65 | 8.35 | 63.16 | 39.94 |
| Post-trained model | LLaMA3-8B-Fin | 8B | 52.65 | 8.39 | **81.00** | **62.23** |

Table 9: Overall performances across all sections. Results are averaged over all the tasks in the corresponding section as shown in Table 2. "—" indicates that the evaluation could not be run due to lack of package support or the requirement for a non-chat format. For instruction following benchmark, we use MT-bench first-turn score.

**Superiority of LLaMA3-8B-Fin.** Table 9 shows the overall performance across both general and in-domain capacities. In general, LLaMA3-8B-Fin outperforms other baselines, including much larger general LLMs and even large LLMs specifically designed for the financial domain, all while maintaining strong general capacities. We give detailed observations below.

**(1) LLaMA3-8B-Fin shows the greatest improvement on seen task.** We observe a huge improvement on seen tasks, approximately 20%. This is expected as our SFT has similar tasks. What is surprising, however, is that our model outperforms the carefully designed financial LLM, Palmyra, by a large margin. It highlights the power of targeted training in LLMs and reinforces the idea that post-training not only boosts performance but can make even smaller models exceptionally well-suited for their specialized areas.

**(2) LLaMA3-8B-Fin also improves significantly on unseen tasks.** Despite having only *2 seen task types* in our SFT training data, we still achieve around a 10% improvement on a diverse set of unseen tasks. Moreover, while the large domain-specific LLM manages to outperform its same-size counterpart (LLaMA3.1-Instruct-70B), our model demonstrates further significant improvement. This suggests that even with limited in-domain seen data, the model can transfer its learning to unseen tasks. It highlights the importance of our well-designed training recipe, ensuring the model retains general knowledge while adapting to domain-specific needs.

**(3) LLaMA3-8B-Fin maintains general learned skills in its base model.** A key consideration in post-training is whether the model retains previously learned general skills. Our model performs similarly to the base model (i.e., LLaMA4-Instruct-8B) on both standard benchmarks and instruction-following tasks. This demonstrates that our replay mechanism effectively preserves general knowledge and stage knowledge. This aspect is often overlooked by domain-specific practitioners (we can see the large domain-specific LLMs (Palmyra-Fin-32K) suffer from severe forgetting on the standard and instruction-following benchmark, compared to its same-size counterpart). While we do note that our model performs worse than some larger models, this is understandable. We anticipate that our training recipe can similarly extend the capabilities of larger LLMs.

## 9 CONCLUSION

Post-training has been widely used in the community to adapt the LLMs, yet a comprehensive analysis remains lacking. In this work, we provide such a timely analysis and propose an effective training recipe, Joint-and-Replay post-training, based on the insights gained from our study. We demonstrate significant improvements in the financial domain, a critical and widely studied area. Notably, we demonstrate that "expanding the web" of an LLM is not only achievable but also highly effective. Our results show that a smaller, domain-specialized LLM can surpass the performance of much larger models. This opens up the exciting possibility that, with the right training recipe, smaller yet better, specialized models can be developed. In the future, we plan to explore a diverse set of domains and expand our analysis to the RLHF stage.

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

## A HYPER-PARAMETERS

**Chat-format.** Chat-format is important for chat-based LLMs. Since our evaluation is not LLM-as-a-judge (except for MT-bench), our results can be sensitive to the chat-format. To prevent bias, we employ both chat-format and non-chat-format for all experiments and report the better result between the two options.

**LLM Hyperparameters**. We set the max length to 8K and pack the samples to make full use of the length capacity. We stop training when the performance on held-out evaluation increases by less than 1 (typically, DAPT stops around 30K steps and SFT around 11K steps). The decoding temperature is set to 0.0 for deterministic outputs. The learning rate is 5e-6 for SFT and joint training, and 1e-5 for DAPT. The warmup ratio is set to 0.1, and gradient checkpointing is applied. All experiments are conducted on 16 A100 40G GPUs.

## B INDIVIDUAL RESULTS

Table 9 already showed the average results across general and in-domain capacities. In this section, we present the individual results for all tasks. Table 10 shows the individual results for in-domain tasks. We observe that LLaMA3-8B-Fin outperforms the baseline on 6 out of 12 tasks, with 4 of those being unseen tasks. It is important to note that many of the baseline LLMs are much larger than our model. Compared to our base model (LLaMA3-Instruct-8B), we perform better on all tasks except OpenQA and stock movement prediction, where in OpenQA we are only slightly behind (by less than 1%). These results indicate that our approach is highly effective. We also notice that for Tabular QA, larger models significantly outperform smaller models, including our base. This suggests that our base model is not strong with tabular data, and naturally we also perform worse on this task. We anticipate that increasing the model size could help improve performance in this particular task.

| Category | Model | Size | Sent Analysis | NER | Rumour Detect | Summ | Fraud Detect | Distress Ident | Claim Analy | ESG Classify | Open QA | Stock Pred | Credit Scoring | Tabular QA |
|---|---|---|---|---|---|---|---|---|---|---|---|---|---|---|
| Domain-specific LLM | Palmyra-Fin-32K | 70B | 0.6737 | 0.5429 | 0.6260 | **0.2751** | 0.4378 | **0.9554** | 0.4924 | 0.3967 | 0.2375 | 0.5474 | **0.5826** | 0.5145 |
| | GPT-4o | N/A | 0.7287 | 0.4302 | 0.7380 | **0.2703** | 0.3827 | 0.7399 | 0.8915 | **0.4567** | **0.2744** | 0.5344 | 0.5719 | 0.6857 |
| General LLMs | LLaMA3.1-instruct | 70B | 0.7081 | 0.4626 | 0.8220 | 0.2657 | 0.1407 | 0.7874 | 0.0228 | 0.4167 | 0.2630 | 0.5531 | 0.4933 | **0.6966** |
| | Mistral-Nemo-instr | 12B | 0.6395 | 0.4984 | 0.8520 | 0.2509 | 0.5396 | 0.3509 | 0.4446 | 0.3267 | 0.2384 | 0.5444 | 0.4479 | 0.5266 |
| | LLaMA3.1-instruct | 8B | 0.6561 | 0.5122 | 0.8420 | 0.2415 | 0.1343 | 0.2317 | 0.1007 | 0.3600 | 0.2010 | 0.5365 | 0.3568 | 0.5506 |
| | Phi-3.5-mini-instruct | 3.8B | 0.6862 | 0.3937 | 0.7540 | **0.2775** | 0.6480 | 0.5576 | 0.5038 | 0.3833 | 0.2108 | 0.4195 | 0.4571 | 0.5097 |
| Post-train base model | LLaMA3-instruct | 8B | 0.6920 | 0.4503 | 0.8260 | 0.2371 | 0.2432 | 0.0872 | 0.4842 | 0.3633 | 0.2436 | **0.5567** | 0.4828 | 0.5342 |
| Post-trained model | LLaMA3-8B-Fin | 8B | **0.8383** | **0.7251** | **0.8620** | 0.2721 | **0.7687** | 0.9243 | **0.9674** | 0.3933 | 0.2338 | 0.5362 | 0.5645 | 0.5454 |

Table 10: Individual results for in-domain capacities. Seen task types are highlighted in grey

Table 11 shows the individual results for general capacities. For general capacities, it is expected that larger models outperform us, as our base model is much smaller. The main focus here is to compare our model with the base model, LLaMA3-Instruct-8B. The results are mixed, and the overall average (as shown in Table 9) is quite similar. This suggests that our model exhibits little to no forgetting.

| Category | Model | Size | MMLU | AI2 ARC | PIQA | Social IQA | GSM8K | MathQA | Trivia QA | NQ Open | Hella swag | Wino grande | Openbook QA | MT Bench |
|---|---|---|---|---|---|---|---|---|---|---|---|---|---|---|
| Domain-specific LLM | Palmyra-Fin-32K | 70B | 0.7708 | 0.7734 | 0.8166 | 0.5133 | 0.7407 | 0.5152 | 0.5228 | 0.1114 | 0.6484 | 0.7388 | 0.2740 | 6.5156 |
| | GPT-4o | N/A | | | | | — | | | | | | | |
| General LLMs | LLaMA3.1-instruct | 70B | 0.8219 | 0.7875 | 0.8324 | 0.5123 | 0.5572 | 0.5578 | 0.7071 | 0.1936 | 0.6521 | 0.7916 | 0.3740 | 9.1438 |
| | Mistral-Nemo-instr | 12B | 0.6594 | 0.7382 | 0.8107 | 0.5154 | 0.3086 | 0.393 | 0.5868 | 0.1271 | 0.6329 | 0.7498 | 0.372 | 8.7000 |
| | LLaMA3.1-instruct | 8B | 0.6775 | 0.7196 | 0.7998 | 0.4928 | 0.2563 | 0.3943 | 0.5178 | 0.1789 | 0.5916 | 0.7411 | 0.3380 | 8.3250 |
| | Phi-3.5-mini-instruct | 3.8B | 0.6851 | 0.7604 | 0.8020 | 0.5742 | 0.6725 | 0.4127 | 0.3650 | 0.1089 | 0.5891 | 0.7474 | 0.3800 | — |
| Post-train base model | LLaMA3-instruct | 8B | 0.6389 | 0.7215 | 0.7835 | 0.4872 | 0.3351 | 0.4204 | 0.5105 | 0.1507 | 0.5765 | 0.7190 | 0.3380 | 8.3500 |
| Post-trained model | LLaMA3-8B-Fin | 8B | 0.6186 | 0.7224 | 0.8020 | 0.4980 | 0.4079 | 0.4137 | 0.4886 | 0.1460 | 0.6071 | 0.7253 | 0.3620 | 8.3875 |

Table 11: Individual results for general capacities.

