# OpenReview forum: "Expanding the Web, Smaller Is Better: A Comprehensive Study in Post-training"
_ICLR.cc/2025/Conference — ICLR 2025 Conference Withdrawn Submission_

### Official Review · Reviewer_TaPq · 2024-10-27

**Soundness:** 2
**Presentation:** 3
**Contribution:** 2
**Rating:** 3
**Confidence:** 4

**Summary:**

This paper mainly analyzes and discusses three significant issues within the post-training phase, including analysis the primary functions of DAPT and SFT, methods to alleviate catastrophic forgetting in the continuous learning process of LLMs, and evaluation of LLMs in both general and specific domains. This work answers the above three questions through experiments. Additionally, they propose a Joint-and-Replay training method to address the problem of catastrophic forgetting.

**Strengths:**

1. This paper discusses training methods for the application of LLMs in a specific domain and highlights three key questions that have great practical value in the post-training phase.
2. This paper is well-organized, featuring concise and clear sentences that facilitate a clear understanding of the core ideas. Furthermore, the figures and tables are well-crafted, effectively presenting the results of the experiments.

**Weaknesses:**

1. The training techniques mentioned in the paper, such as masking the content of instructions in SFT and using a replay strategy to mitigate forgetting [1][2][3], are commonly employed training skills in the LLMs field. Even the proposed Joint-and-Replay training method in this paper is also a commonly used training skill, lacking significant distinctions or standout features compared to existing methods. Perhaps the author should highlight the differences between the proposed methods and existing works in the paper.

[1]Llama 2: Open Foundation and Fine-Tuned Chat Models

[2] Qwen2 Technical Report

[3]Fine-tuned Language Models are Continual Learners

2. While the author raises three crucial issues within this domain, the core conclusions drawn from these three questions do not present any remarkable insights. The first two questions have been extensively explored in previous literature[4][5][6][7]. Furthermore, the evaluation methods discussed in the post-training stage still adhere to standard procedures without introducing novel evaluation approaches. After further post-training to enhance the model's capabilities in specific domains, can a new evaluation method be introduced to dynamically evaluate the model's performance in specific tasks and general domain knowledge? I suggest the authors explore other innovative evaluation methods from these perspectives in the paper.

[4] LIMA: Less Is More for Alignment

[5] Does Fine-Tuning LLMs on New Knowledge Encourage Hallucinations?

[6] Analyzing the Forgetting Problem in Pretrain-Finetuning of Open-Domain Dialogue Response Models

[7] Simple and Scalable Strategies to Continually Pre-train Large Language Models

3. The authors should focus on **introducing their proposed methods in the Method section**, rather than extensively on basic training techniques like SFT and pre-training in section 3.1.

**Questions:**

**Joint Training Details:** The paper mentions joint training of DAPT and SFT. How did you set the training ratio? Did you experiment with different ratios, such as an even split between DAPT and SFT, or a higher proportion for DAPT? Could you provide more details on the joint training process, such as training time, learning rate, etc.?

**Evaluation of Unseen Tasks:** The paper mentions that unseen tasks are primarily from the financial domain. How did you define unseen tasks? Did you attempt using data from other domains as unseen tasks? Could you provide more details on the evaluation of unseen tasks, such as evaluation metrics, task types, etc.?

---

### Official Review · Reviewer_MAEb · 2024-11-03

**Soundness:** 3
**Presentation:** 3
**Contribution:** 2
**Rating:** 5
**Confidence:** 3

**Summary:**

This paper explored the post-training of language models to adapt them for domain-specific tasks. In particular, the paper discusses Domain-Adaptive Pre-Training (DAPT), and Supervised Finetuning (SFT) for pretrained and chat-tuned models (LLaMA3 8B). The authors evaluated the proposed post-training scheme on the financial domain, and showed that their model can outperform a SoTA finance-specific language model on a certain set of tasks. The authors also showed that joint DAPT and SFT where the model is trained jointly on text-token prediction on raw text and instruction following provide the best performance, and results in the least amount of reduction in the general capabilities of the pretrained model (i.e., the least forgetting).

**Strengths:**

- The paper considered the important problem of post-training of language models to adapt them for domain-specific tasks
- The paper presented different design choices, and proposed the use of joint training with replay to achieve best performance
- The approach was applied to obtain a state-of-the-art language model for the financial domain

**Weaknesses:**

- The selection of different datasets seemed arbitrary and confusing. Aqua, math, GSM-8k, and other evaluation datasets were included during pretraining, which defeats the purpose of evaluation. I would have preferred to use generic pretraining datasets such as Wikitext/C4/PILE/FineWeb etc.
- Any description of datasets, and the associated choices were missing from the paper, making it hard to understand the reason for those choices. This made it hard to understand the significance and the reliability of the obtained results. The authors only mentioned filtering of URLs from FineWeb without any further details. The inclusion of details about each of the datasets and the reasons for their inclusion is important to be specified.
- The paper just explored SFT, without considering distillation as a potential remedy for forgetting. I would be keen to understand the impact of distillation, or even just generation of the target documents from the model.
- The reliability of the comparison with the other SoTA model (Palmyra) is unclear. No description of what that model is trained on is provided. It would be important to elaborate on the comparison.

**Questions:**

- Was the answer for the dataset such as GSM-8k also included during DAPT (table 2a)?
- What kind of documents from FineWeb were filtered? Mentioning that they were selected based on URLs is very vague
- Why did the authors decide to include evaluation datasets instead of regular pretraining datasets such as PILE/C4/FineWeb?
- Why was the comparison against the other SoTA financial model (Palmyra) fair? Was the model trained and exposed to the same datasets that the authors used for finetuning?

---

### Official Review · Reviewer_Wpsy · 2024-11-04

**Soundness:** 2
**Presentation:** 3
**Contribution:** 1
**Rating:** 3
**Confidence:** 4

**Summary:**

The paper aims to focus on post-training for LLMs, specifically on the financial domain. It investigates Domain-Adaptive Pre-Training (DAPT) and Supervised Fine-Tuning (SFT) in building a finance-specific model, LLaMA3-8B-Fin, using a proposed "Joint-and-Replay" approach. This methodology aims to enhance domain-specific knowledge retention while maintaining general capabilities. The evaluation includes both general benchmarks and finance-specific tasks, with findings suggesting that LLaMA3-8B-Fin achieves competitive performance on finance tasks compared to larger models.

**Strengths:**

1. Focused application on finance. The paper’s focus on developing a LLM for finance and highlights practical insights for domain-specific adaptation.
2. The Joint-and-Replay approach is straightforward and could offer a useful recipe for practitioners aiming to balance domain-specific and general knowledge in smaller LLMs.
3. Experiments were conducted comprehensively on sufficient number of existing evaluation datasets.

**Weaknesses:**

1. Insufficient support for broad claims on post-training.
    - The paper asserts a broad investigation into the entire post-training stage for LLMs but does not sufficiently review or acknowledge recent advancements in this area. Particularly, the claim that "most approaches merely involve additional pre-training on specialized data or rely on the traditional LLM framework where a single pre-training stage is followed by task-specific fine-tuning via classifiers" (lines 77–83) fails to account for significant contributions in post-training, including RLHF which is a very important stage in post-training.
    - Extending from SFT, research in post-training has also focused on self-training methods like RFT [1], STaR [2-3], ReST^EM [4] and self-reward [5]. There have also been works that aim to unify SFT with RLHF [6] and numerous works studying SFT/DPO/PPO. The technical reports of Llama-3, for example, dedicated substantial amount of pages to discuss their post-training techniques. The paper’s claim of a comprehensive study does not seem well-supported to me.
   - The claim of scope for this paper should focus on LLM for finance instead of post-training.

2. Unclear significance in general benchmark evaluation. The general benchmark results show that LLaMA3-8B-Fin does not degrade compared to LLaMA3-8B-Instruct but underperforms other LLMs. Since domain-specific models like Palmyra-Fin-32K also maintain performance on standard benchmarks, this result lacks a clear takeaway.

3. The results in Table 8, showing full fine-tuning outperforming LoRA, are expected. It’s widely understood that full fine-tuning generally yields better results when computational resources are sufficient. This section does not add novelty, as many models (e.g., LLaMA) already favor full fine-tuning for optimal performance.

4. Section 7 on evaluation lacks significance in contribution, given the extensive existing work on LLM evaluation.


[1] Scaling relationship on learning mathematical reasoning with large language models

[2] STaR: Self-Taught Reasoner

[3] V-star: Training verifiers for self-taught reasoners

[4] Beyond human data: Scaling self-training for problem-solving with language models

[5] Self-rewarding language models

[6] Intuitive Fine-Tuning: Towards Unifying SFT and RLHF into a Single Process

**Questions:**

See above in weaknesses.

---

### Note · Authors · 2024-12-03

I have read and agree with the venue's withdrawal policy on behalf of myself and my co-authors.